# Protection of Cholinergic Neurons against Zinc Toxicity by Glial Cells in Thiamine-Deficient Media

**DOI:** 10.3390/ijms222413337

**Published:** 2021-12-11

**Authors:** Sylwia Gul-Hinc, Anna Michno, Marlena Zyśk, Andrzej Szutowicz, Agnieszka Jankowska-Kulawy, Anna Ronowska

**Affiliations:** 1Department of Laboratory Medicine, Medical University of Gdansk, 80-210 Gdansk, Poland; sylwia.gul-hinc@gumed.edu.pl (S.G.-H.); anna.michno@gumed.edu.pl (A.M.); agnieszka.jankowska-kulawy@gumed.edu.pl (A.S.); marlena.zysk@gumed.edu.pl (A.J.-K.); 2Department of Molecular Medicine, Medical University of Gdansk, 80-210 Gdansk, Poland; andrzej.szutowicz@gumed.edu.pl

**Keywords:** zinc toxicity, acetyl-CoA, energy metabolism, thiamine pyrophosphate

## Abstract

Brain pathologies evoked by thiamine deficiency can be aggravated by mild zinc excess. Cholinergic neurons are the most susceptible to such cytotoxic signals. Sub-toxic zinc excess aggravates the injury of neuronal SN56 cholinergic cells under mild thiamine deficiency. The excessive cell loss is caused by Zn interference with acetyl-CoA metabolism. The aim of this work was to investigate whether and how astroglial C6 cells alleviated the neurotoxicity of Zn to cultured SN56 cells in thiamine-deficient media. Low Zn concentrations did not affect astroglial C6 and primary glial cell viability in thiamine-deficient conditions. Additionally, parameters of energy metabolism were not significantly changed. Amprolium (a competitive inhibitor of thiamine uptake) augmented thiamine pyrophosphate deficits in cells, while co-treatment with Zn enhanced the toxic effect on acetyl-CoA metabolism. SN56 cholinergic neuronal cells were more susceptible to these combined insults than C6 and primary glial cells, which affected pyruvate dehydrogenase activity and the acetyl-CoA level. A co-culture of SN56 neurons with astroglial cells in thiamine-deficient medium eliminated Zn-evoked neuronal loss. These data indicate that astroglial cells protect neurons against Zn and thiamine deficiency neurotoxicity by preserving the acetyl-CoA level.

## 1. Introduction

Neurons in the brain have high energy requirements that make them particularly susceptible to metabolic stress. The impairment of glucose, oxygen or lactate provision causes energy hypometabolism, inducing several neurodegenerative disorders, e.g., Alzheimer’s disease (AD) [1]. Energy failures result from a direct or an indirect inhibition of the tricarboxylic acid cycle and the electron transport chain by multiple neurodegenerative signals, such as an excess of Zn, Al or NO, hypoxia, hypoglycemia, amyloid β-protein and hyperphospho-tau oligomerizations [2]. These factors inhibit pyruvate dehydrogenase (PDH), α-ketoglutarate dehydrogenase complexes (KDH), and aconitase activities in mitochondria. Furthermore, the suppression of PDH activity decreases acetyl-CoA synthesis and utilization in the tricarboxylic acid (TCA) cycle [2,3]. On the other hand, a decreased acetyl-CoA level reduces cytoplasmic acetylcholine production, resulting in poor neurotransmission and decreased cell viability [2,3,4,5]. Moreover, the susceptibility of SN56 cholinergic cells to cytotoxic insults increases with the cholinergic phenotype [4,5].

Neuronal energy metabolism is affected by thiamine deficiency (TD) resulting from a thiamine-poor diet [6]. TD is identified in patients (the majority of them are chronic alcoholics) suffering from hyperemesis gravidarum, intestinal obstruction or malignancy [6].

One neuropsychiatric disorder arising from TD is called Wernicke’s encephalopathy [7]. Wernicke’s encephalopathy can be reversed if a patient is treated with adequate doses of thiamine within the first 48–72 h of the first clinical symptoms appearing [6]. These symptoms are caused by a disturbance in cholinergic neurotransmission. The acetylcholine (ACh) decreases, and choline acetyltransferase (ChAT) inhibition occurs in areas containing cholinergic neurons in TD brains. This preferential loss of cholinergic neurons is also one of the hallmarks in the progression of Alzheimer’s disease (AD) [2]. Therefore, our working hypothesis is that energy disturbances in AD can by triggered by thiamine deficiency. Thiamine is phosphorylated to a pyrophosphate of thiamine (TPP) by thiamine pyrophosphokinase [7]. TPP is most highly concentrated in the human brain. Thus, thiamine deficiency leads to TPP deficits. TPP is the cofactor for the E1 subunit of PDH and KDH [8]. Moreover, asymptomatic forms of TPP deficiency have been frequently reported in populations of elderly people. This symptom is considered to be one that worsens the course of AD [9,10]. Therefore, there are similarities between the cognitive dysfunction in AD and TD patients. The most important similarity is the inhibition of glucose metabolism [10]. Indeed, postmortem studies on the brain cortex of patients who died from AD show the inhibition of PDH activity [11]. However, TPP deficiency affects oxidative metabolism and function not only in neurons but in all types of glial cells. The data showed that there was an inhibition of PDH activity that caused a decrease in the metabolic flux in the synaptosomes of rats fed with pyrithamine, which is an analog of thiamine inducing thiamine deficiency [12]. The same observation was noticed in SN56 cholinergic cells cultured with amprolium, a competitive inhibitor of thiamine transport. The intracellular TPP deficit led to a decrease in acetyl-CoA and ACh synthesis due to PDH activity inhibition, which increased the mortality of the cultured neurons [2,4].

The next cytotoxic insult that has been proven to limit acetyl-CoA metabolism is an excessive release of Zn from glutamatergic terminals [3,13,14]. This phenomenon is considered to be a primary neurodegenerative signal. Under physiological conditions, Zn co-released with glutamate is quickly cleared from the synaptic cleft by astrocytes and postsynaptic neurons [13]. However, under pathological conditions, there is an excessive release of Zn-glutamate from depolarized glutamatergic presynaptic terminals, which increases the free Zn concentration (Figure 1) [15]. This is a consequence of neurotoxic factor exposure, mechanical trauma, ischemia, or TD [5]. The excess of free (unbound) Zn^2+^ is accumulated in depolarized postsynaptic neurons, where it inhibits PDH activity, leading to a decrease in the acetyl-CoA level. Additionally, KDH and aconitase and NADP-isocitrate dehydrogenase (ICDH-NADP) activities were inhibited by elevated Zn^2+^ (Figure 1) [16,17]. This is the main reason for excessive neuronal death [3]. In turn, the aberrant redistribution of Zn in postsynaptic cellular compartments triggers neurotoxic events [13]. However, TD aggravates the Zn^2+^ neurotoxicity by increasing the intracellular accumulation of the cation [5]. Moreover, the susceptibility of the neurons to Zn and other cytotoxic signals may differ depending on their phenotype. It has been shown that differentiated septal SN56 cholinergic cells displayed greater susceptibility to Zn excess and TD than nondifferentiated ones (Figure 1) [2,5]. Thus, maintenance of the pyruvate-acetyl-CoA(mit)-acetyl-CoA(cyt)-ACh pathway constitutes a functional unit in cholinergic neurons.

However, earlier experiments did not take the impact of cytotoxic factors on energy metabolism in glial cells into consideration. For instance, astrocytes play an important role in maintaining neuronal homeostasis, providing lactate as a complementary energy substrate for neurons [18]. They protect neurons against different neurotoxic conditions, including heavy metal intoxication, hypoxia, and other neurodegenerative conditions [19]. The impairment of oxidative metabolism in astrocytes with thiamine deficiency may induce lactate and glutamate accumulation, causing acidosis and the excitotoxic stimulation of glutamatergic neurons [20,21]. To the best of our knowledge, there are no data showing whether astroglial cells are capable of protecting neuronal cells against Zn toxicity in TD. If this is the case, they could clear Zn from the neuronal vicinity, thereby protecting neurons against Zn overload. However, it is not known whether astroglial cells respond to marginal Zn excess in thiamine deficiency, especially in terms of their energy metabolism and viability. This knowledge would give us a more detailed answer about the energy metabolism in the brain. Therefore, the aim of this work was to investigate whether such protective astroglia-neuron interactions might take place under Zn overload in thiamine-deficient conditions. The data presented here indicate that this might be the case.

## 2. Results

### 2.1. Concentration-Dependent Effects of Amprolium and Zn on Thiamine Pyrophosphate (TPP), Intracellular Zn Content, and Viability of C6 Cells and Primary Glial Cells

The TPP level was about 73 pmols/mg protein in C6 cells grown in medium supplemented with 0.009 mmol/L thiamine (Figure 1a) [5]. The addition of amprolium up to 10 mmol/L to a medium supplemented with 0.009 mmol/L thiamine brought about no significant alterations in the TPP level of the tested cells (Figure 1a) [5]. Amprolium is a competitive inhibitor of the thiamine transporter and is used to aggravate a TPP deficit [21].

C6 and primary glial cells grown in a thiamine-deficient medium displayed TPP levels equal to about 60 pmol/mg protein in control (Figure 1a). In these conditions, amprolium in 5 and 10 mmol/L concentrations caused about a decrease of roughly 45% of TPP in C6 cells (Figure 1a). The increase in amprolium to 20 mmol/L brought about an 80% suppression of the TPP level (Figure 1a). Zn added to the cell culture in amounts up to 0.20 mmol/L did not significantly decrease the TPP intracellular level when the cells were grown in a thiamine-supplemented medium (Figure 1b). In turn, there was a 28% decrease in the intracellular TPP level at 0.20 mmol/L Zn in the cells cultivated in a thiamine-deficient medium (Figure 1b).

The control intracellular Zn level in C6 cells was equal to 0.98 nmol/mg protein (Figure 1c,d). The addition of amprolium did not affect the Zn level (Figure 1c). To accumulate 3.0 nmol/mg protein of Zn, C6 cells required the addition of 0.15 mmol/L of this cation to the cell culture (Figure 1d) [5].

The viability of C6 cells and primary glial cells was not decreased by amprolium when cells were cultured in a thiamine-supplemented medium (Figure 2a,c). However, in C6 cells, 10 mmol/L amprolium resulted in a 20% drop in the C6 cell number (Figure 2a) and a 25% decrease in the primary glial cell viability (Figure 2c). The increase in amprolium to 20 mmol/L brought about an 80% suppression of the cell count in C6 cultured in a thiamine-deficient medium (Figure 2a).

Zn at a 0.20 mmol/L concentration caused a roughly 25% loss of C6 cell count (Figure 2b). Furthermore, an increase in Zn to 0.30 mmol/L caused an instant loss of C6 cells independently of the thiamine presence in the culture media (Figure 2b). The viability of the primary glial cells measured using the MTT reduction test was not significantly changed by Zn, irrespective of the thiamine presence in the medium (Figure 2d).

Therefore, to examine the combined effect of TD and Zn on C6 cells, 5 and 10 mmol/L amprolium and 0.15 mmol/L Zn were chosen for further experiments because they did not significantly decrease cellular viability. For primary glial cells, 10 mmol/L amprolium and 0.20 mmol/L Zn were chosen.

### 2.2. Concentration-Dependent Effects of Amprolium and Zn on Enzymes of Energy Metabolism and Acetyl-CoA Level in C6 Cells

Standard PDH assays are performed at saturating concentrations of substrates and cofactors including TPP. Therefore, this reflects the amount of active enzyme complex in the cells [5]. Here, we additionally used the modified assay medium without exogenous TPP to assess the fractional saturation of PDH with TPP and putative metabolic flux through this step in cells in situ [22]. The control activities of PDH measured with TPP added to the assay medium were over two times higher than those measured without this cofactor (Figure 3a,b). The PDH activity in C6 cultivated in thiamine-supplemented medium was 30% higher than the one found in the cells grown without thiamine and was resistant to a Zn addition (Figure 3b). Amprolium in concentrations of 10 mmol/L brought about no alterations in PDH activity but reduced the acetyl-CoA level by about 35% in C6 cells cultured in thiamine-deficient MEM (Figure 3a,c). In the same conditions, 0.15 mmol/L Zn did not alter the enzyme activity or the acetyl-CoA levels (Figure 3b,d). Higher concentrations of 0.20 mmol/L Zn reduced the PDH activity and the acetyl-CoA level, respectively, by 50 and 75% (Figure 3b,d).

Aconitase and ICDH-NADP activities in C6 cells were not significantly inhibited by 10 mM amprolium (Figure 4a,c) or 0.15 mM Zn (Figure 4b,d). Higher concentrations of 20 mmol/L amprolium and 0.20 mmol/L Zn resulted in a 73 and 50% inhibition of aconitase (Figure 4a,b) and no major alterations in ICDH-NADP activities, respectively (Figure 4c,d).

Amprolium at 10 mmol/L and Zn at 0.15 mmol/L were chosen for studies of their combined toxicity on C6 and primary glial cells on the basis of the data presented in Figure 1, Figure 2, Figure 3 and Figure 4. On the other hand, 5 mmol/L amprolium and 0.10 mmol/L Zn alone were harmful for SN56 cholinergic neuronal cells, and this cytotoxicity was aggravated by their combined use [5]. On the other hand, the same concentrations of the compounds were not toxic for C6 cells cultivated in a thiamine-deficient medium (Figure 2, Figure 3, Figure 4 and Figure 5). Therefore, the latter concentrations of amprolium and Zn were used to study the putative neuroprotective effects of C6 astroglial cells on SN56 cells.

### 2.3. Combined Cytotoxic Effects of Zn and Thiamine Deficits on C6 Astroglial and SN56 Cholinergic Neuronal Cells. The Effect of C6 Inserts on SN56 Cells

#### 2.3.1. The Effects of Zn, Amprolium, and C6 Inserts on Intracellular TPP and Zn Levels

There was a roughly 35% reduction in the intracellular TPP content in C6 astroglial or SN56 cells cultured in a thiamine-deficient medium with 5 mmol/L amprolium alone (Figure 5a,b). The 0.10 mmol/L Zn alone brought about a 25% reduction in TPP in SN56 cells (Figure 5b), and none in C6 cells. Zn did not enhance the suppressive effects of amprolium on the TPP levels in C6 (Figure 5a), but it augmented its deficit to 50% in SN56 cells (Figure 5b) [5]. To study the interaction between C6 and SN56 cells, we co-cultured SN56 with C6 seeded on cell culture inserts (with a 0.40-micrometer pore, 3-centimeter diameter). The intracellular TPP level was not changed in SN56 cells cultured with C6 inserts against the control (Figure 5b). However, the presence of C6 inserts prevented a TPP decrease in the SN56 cells exposed simultaneously to amprolium and Zn (Figure 5b).

The intracellular Zn level was not changed by the addition of 5 and 10 mmol/L amprolium in C6 cells (Figure 5c). Nonetheless, it was elevated over four times by 0.15 mmol/L Zn and 5 mmol/L amprolium (Figure 5c). Amprolium and C6 inserts alone did not change the intracellular Zn content in the SN56 cells (Figure 5d). The exposure of the SN56 neuronal cells to 0.10 mmol/L Zn and 5 mmol/L amprolium resulted in a partially additive increase in intraneuronal Zn content from 1.74 to 2.49 nmol/mg protein (Figure 5d). Inserts with C6 cells prevented Zn elevation in SN56 cells (Figure 5d).

#### 2.3.2. The Effects of Zn, Amprolium, and C6 Inserts on PDH Activity, Acetyl-CoA Level, and the Total Cell Number

Amprolium (5 mmol/L) as a sole cytotoxic factor did not affect the PDH activities in the C6 and SN56 cells (Figure 6a,b). Zn alone did not inhibit the enzyme’s activity in C6 cells (Figure 6a) but there was a 50% inhibition of PDH in SN56 cells (Figure 6b). The combined addition of 10 mmol/L amprolium and 0.15 mmol/L Zn brought about no significant decrease in PDH activity in the C6 cells (Figure 6a). On the other hand, 5 mmol/L amprolium and 0.10 mmol/L Zn caused a total inhibition of PDH activity in the SN65 cells (Figure 6b). The insert of C6 cells did not alter the basal PDH activity in SN56 but partially prevented amprolium+Zn-evoked inhibition (Figure 6b). Zn (0.15 mmol/L) or amprolium (10 mmol/L) alone caused a roughly 15% decrease in the acetyl-CoA level in a thiamine-deficient medium in C6 cells (Figure 6c). The combined use of these factors augmented acetyl-CoA suppression to 50% (Figure 6c). The acetyl-CoA level was reduced by about 30% in the SN56 neuronal cells cultured with 5 mmol/L amprolium or 0.10 mmol/L Zn alone in a thiamine-deficient medium (Figure 6d). The joint addition of these compounds caused a 63% drop in the neuronal acetyl-CoA level. The inserts of C6 cells reversed this suppressive effect (Figure 6d).

The total number of C6 cells was not significantly decreased by a separate treatment of amprolium or Zn. However, a combined culture with these compounds brought about a 34% drop in the C6 cell number (Figure 6e). Single additions of amprolium or Zn to cultured SN56 cells caused 35 and 45% decreases in their number, respectively (Figure 6f). The combined use of Zn and amprolium aggravated the SN56 cell number depletion by 80% (Figure 6f). Inserts of C6 cells brought back the SN56 cell number to 85% of the control level (Figure 6f).

#### 2.3.3. The Effects of Zn and Amprolium and C6 Inserts on the Activities of Selected Enzymes of the TCA Cycle

The chronic exposure of C6 cells to a thiamine-deficient medium at 0.15 mmol/L Zn or 10 mmol/L amprolium alone resulted in a 25 and 30% inhibition of aconitase activity (Figure 7a). The enzyme’s activity was inhibited by 52% by a combined use of these factors (Figure 7a). There were no alterations in ICDH-NADP activities (Figure 7c). The aconitase activity in neuronal SN56 cells was not affected by 5 mmol/L amprolium.

The enzyme was inhibited by 66% with 0.10 mmol/L Zn (Figure 7b). In turn, amprolium+Zn inhibited aconitase activity by 86% in the SN56 cells (Figure 7b). Inserts of C6 cells did not alter aconitase activity. They reversed the inhibitory effect of amprolium+Zn on aconitase activity by 50% (Figure 7b). Amprolium had no effect on ICDH-NADP activity in neuronal SN56 cells (Figure 7d). On the other hand, Zn alone or Zn+amprolium inhibited enzyme activity by 85 and 75%, respectively (Figure 7d). Inserts of C6 cells partially reversed this inhibition in SN56 neuronal cells (Figure 7d).

### 2.4. Combined Effects of Amprolium and Zn on Metabolic Parameters of Primary Glial Cells

Amprolium in 10 mmol/L concentrations added to a medium supplemented with 0.009 mmol/L thiamine brought about no significant alterations in the TPP level.

Primary glial cells grown in thiamine-deficient medium displayed TPP levels equal to about 60 pmol/mg protein in the control (Figure 8a). In these conditions, 10 mmol/L amprolium caused a 24% decrease in the intracellular TPP content (Figure 8a). Separate additions of 10 mmol/L amprolium or 0.20 mmol/L Zn caused 24 and 20% decreases in the intracellular TPP content (Figure 8a). The suppressive effects on the TPP content were augmented by the combined application of the tested compounds (Figure 8a).

The control intracellular Zn level in primary glial cells was equal to 2.09 nmol/mg protein (Figure 8b). Amprolium did not change the intracellular Zn level (Figure 8b), but this level was elevated to 3.12 nmol/mg protein in cells cultured with 0.20 mmol/L Zn (Figure 8b). A culture with 10 mmol/L amprolium and 0.20 mmol/L Zn caused a further increase in the intracellular Zn level to 3.69 nmol/mg protein (Figure 8b).

The metabolic parameters of primary glial cells were tested here to verify whether the properties of clonal C6 cells may be comparable to those of primary glial cells.

The activity of PDH was 50% lower when measured in the presence of TPP in the assay medium (Figure 8c,d). There was a 20% decrease in PDH activity at 10 mmol/L amprolium in both of the tested assay media (Figure 8c,d). Zn in concentrations of 0.20 mmol/L inhibited the enzyme’s activity by 26% only in the assay without TPP (Figure 8d). Amprolium+Zn caused a 23% inhibition of PDH activity in the presence of TPP in the assay medium (Figure 8c). Yet, amprolium+Zn caused a 35% inhibition of PDH activity without TPP in the assay medium (Figure 8d). The PDH activity inhibition by 10 mmol/L amprolium or 0.20 mmol/L Zn led to a roughly 30% acetyl-CoA level drop (Figure 8e). This effect was aggravated over two times by the combined use of amprolium and Zn (Figure 8e). It caused over a 50% loss of cellular viability measured by the MTT reduction rate (Figure 8f).

## 3. Discussion

There are a number of experimental data presenting the separate, detrimental effects of either Zn excess or thiamine deficiency (TD) on the energy metabolism and survival of brain cells [23]. Our recent study proved that there is an enhancement of Zn accumulation caused by TPP deficiency in cholinergic neuronal cells [5]. This caused a decrease in neuronal viability and neurotransmission by its interaction with acetyl-CoA metabolism. On the other hand, to the best of our knowledge, there are no data on interactions between these pathological signals in glial cells. Since astroglia take an active role in the maintenance of neuronal homeostasis in the brain, knowledge of their metabolism in toxic conditions would be essential to prevent neurodegeneration. Therefore, the presented study investigated the combined effects of minor Zn excess and TD on key steps of energy metabolism in astroglial cells. Then, the influence of astroglial cells on neighboring neuronal SN56 septal cholinergic cells subjected to cytotoxic insults was studied. The obtained results might help to explain the role of astroglia in the early stages of neurodegeneration.

The C6 astroglial cell line was used as model of neuroprotection within this study. Primary astrocyte cultures create difficulties in studying their energy metabolism parameters because of the limitation of their growth. On the other hand, a C6 culture is an isolated system. Yet, it has been demonstrated recently that C6 cells show a similar gene expression to isolated astrocytes [24]. However, both types of cells displayed a similar response and susceptibility to Zn excess. C6 required 0.15 and primary glial cells 0.20 mmol/L Zn in the culture medium to accumulate its intracellular level of about 3.0 nmol/mg protein. This led to a 25% loss of their viability. Furthermore, the presented data indicate that a culture of C6 glial and primary glial cells in a thiamine-deficient medium caused a similar, slight 24–28% decrease in their TPP content. This drop did not significantly affect the growth rate, the acetyl-CoA level, or the aconitase and ICDH-NADP activities in C6 and primary glial cells. This is why such deficits are marginal for them. On the other hand, a drop in TPP caused a 30% inhibition of PDH activity in glial cells. The same was observed in TD cholinergic SN56 cells, which preserved parameters of viability, but PDH activity was inhibited in conditions of marginal thiamine deficiency [5,23]. These findings are compatible with studies on TD rats, which demonstrated no apparent symptoms of TD at 20% decreases in the TPP levels in their brains [25]. This indicates that C6 clonal cells may constitute the in vitro model of TD in astroglial cells. Decreases in the TPP levels in rat’s brains or human serum of more than 50% decreased PDH activity, leading to overt clinical and pathological symptoms [12,26]. However, under moderate TD conditions, neurons and glial cells displayed different susceptibility. For instance, treatment with amprolium alone decreased TPP levels by about 20, 31, and 48%, respectively, in primary glial cells, SN56, and C6 cells. However, there was only reduced viability in SN56 neuronal cells. This indicates that there is a relative resistance of glial cells to moderate TPP deficits. This may be a promising phenomenon for neuroprotection against cytotoxic insults.

Within this study, we used an additional cytotoxic factor: Zn excess. This aggravated the decrease in the TPP level both in C6 and primary glial cells. In consequence, there was a similar suppression of PDH activity and the acetyl-CoA level and a marked decrease in their viability.

The cells also displayed a variable susceptibility to Zn excess. Thus, there was a 50% inhibition of SN56 viability at 0.10 mmol/L Zn, a 15% inhibition of C6 at 0.15 mmol/L, whereas primary glial cells did not lose their viability even at a 0.20 mmol/L Zn concentration. The mortality of SN56 was accompanied by corresponding alterations in PDH activity (Table 1). It is known that the cytotoxic effects of Zn on cultured SN56 neuronal cells appear above 0.10 mmol/L when its concentration exceeds the binding capacity of the serum proteins present in a standard culture medium [14,17]. Therefore, the estimated concentration of ionic Zn in the growth medium is equal to about 0.01 mmol/L in such conditions [17,27]. Therefore, concentrations of free Zn^2+^ in the presence of 0.15 and 0.20 mmol/Zn in the culture medium in the C6 and primary astroglia media are equal to 0.06 and 0.11 mmol/L, respectively. Previous data showed that 0.01 mmol/L Zn^2+^ exerted a significant toxic effect on SN56 in a thiamine-deficient medium (Table 1a). This indicates that a relatively minor excess of Zn^2+^ in the extracellular fluid is able to penetrate plasma membranes and accumulate in the TD neurons. It may result from a several-fold increase in 1,2 L–type Ca channel levels in TD neurons [28,29]. However, glial cells displayed a different capacity for accumulate and buffer Zn^2+^ because they have diverse transporting and complexing systems. In fact, neuronal cells expressing high levels of the ZnT protein and mRNA have a high affinity for voltage gated calcium channels and present a fast Zn^2+^ uptake [30] (Table 1a). Indeed, neuronal cells cultured in TD media with only 0.10 mmol/L Zn accumulated 2.3 nmols/mg protein. On the other hand, glial cells needed higher additions of 0.15 and 0.20 mmol/L Zn to TD culture media to accumulate a similar level of this cation. Thus, astrocytes have an affinity but high capacity for Zn accumulation. Furthermore, they are more resistant to Zn excess than neuronal cells. Therefore, at 0.10 mmol/L Zn, C6 astroglia stayed viable and retained their neuroprotective properties.

Combined applications of amprolium and Zn resulted in similar 48, 32 and 42% deficits of TPP in tested cells (Table 1b). However, PDH activity was inhibited with a different strength. For example, there was total PDH activity inhibition in SN56 neuronal cells by about 40% in both types of glial cells. This indicates that the lack of TPP as the PDH cofactor may not be the only factor responsible for the inhibition of the enzyme activity that leads to viability loss of neuronal cells in cytotoxic conditions (Table 1b). Nevertheless, both neuronal and astroglial cells possess similar mechanisms of thiamine transport and the conversion of thiamine to TPP yielding its similar distribution along all the cellular compartments of the brain [21]. However, the greater sensitivity of SN56 to TD and other cytotoxic insults may be due to the fact that acetyl-CoA is used both for energy production and acetylcholine synthesis. Therefore, cholinergic neurons require much higher PDH activity for acetyl-CoA synthesis. On the other hand, the PDH activity in glial cells was two times lower than in neuronal cells and more resistant to Zn excess. This is compatible with the faster rate of glycolysis and slower pyruvate metabolism in astroglia than in neuronal cells [19,31]. Additionally, the activity of PDH in glial cells was resistant to moderate Zn inhibition, thereby preserving the acetyl-CoA level for energy production in the tricarboxylic acid cycle (Table 1a). This might be explained by the fact that there is an increased level of the S100A6 protein in both primary and clonal astrocytes in response to Zn overload [32]. This may be the reason for the greater resistance of C6 and primary glial cells to isolated zinc excess in thiamine-deficient conditions (Figure 9c). Thus, astroglia may exert neuroprotective effects by taking up Zn excess from the neuronal vicinity [19]. This thesis is supported here by the data, indicating that C6 inserts accumulate higher amounts of Zn than SN56 cells in neurotoxic conditions (Figure 9a–d).

TPP deficits caused by amprolium enhanced the Zn^2+^ uptake into the tested cells. However, the influx was differentiated in the various cell types. Moreover, the PDH activity and acetyl-CoA level depended more heavily on the intracellular Zn level in SN56 than in C6 cells (Figure 9a,b). In this case, aggravated Zn uptake is only crucial for neuronal survival (Figure 9c). Therefore, there is only a significant correlation between the TPP level and Zn accumulation in neuronal cells (Figure 9d). This indicates that the viability of neurons strongly depends on PDH-derived acetyl-CoA. That is why any cytotoxic signal that inhibits PDH activity causes excessive neuronal death. Such a significant dependence was not observed in glial cells. However, only very high Zn concentrations did inhibit PDH activity in glial cells in thiamine-deficient conditions (Figure 1). The mechanism of Zn and amprolium cytotoxicity on PDH is based on the removal of lipoamide from the E2 subunit, whereas amprolium suppressed the E1 subunit by the TPP depletion of its active centre [16]. This indicates the existence of a dual mechanism of PDH susceptibility. Therefore, Zn affects both neurons and astroglia but with a different potency.

Zn also inhibited aconitase activity by the displacement of Fe^2+^ from its active centre [17]. The inhibition of aconitase by Zn in C6 was weaker than in SN56 cells, which may be an additional factor contributing to the higher susceptibility of neuronal cells to Zn excess (Table 1, Figure 9e). The strong inhibition of ICDH-NADP by Zn may be an additional complementary factor hampering only SN56 viability. On the other hand, the lack of ICDH-NADP activity inhibition by Zn in C6 cells may promote their resistance against the accumulation of this cation (Figure 9f). This indicates that astroglial cells may retain their functional capacity in conditions harmful for neuronal cells.

Thanks to these properties, C6 astroglial cells protect the neuronal SN56 cells against TD and Zn toxicity. The neuroprotective effects of C6 astroglial cells may be caused by a decreased Zn accumulation and an increased TPP level in neuronal SN56 cells, and by a higher Zn^2+^ uptake. Thereby, C6 cells prevented the inhibition of PDH activity and acetyl-CoA synthesis, yielding the maintenance of neuronal viability.

## 4. Materials and Methods

### 4.1. Cell Cultures

The glioblastoma C6 cell line (RRID: CVCL 0194) and cholinergic neuroblastoma SN56.B5G4 cells (RRID: CVCL 4456) were used as models of Zn excess and TPP deficit cytotoxicity [5]. The SN56 cells were checked for their cholinergic phenotype routinely once a month by an acetylcholine level measurement, which should be no less than 30 pmols/mg protein in highly differentiated cholinergic cells. The C6 cells were verified by binding with CD11b (Invitrogen, Waltham, MA, USA) antibody or RAN (Invitrogen, Waltham, MA, USA), respectively, with the use of the MACS system (Miltenyi Biotec GmbH, Bergish-Gladbach, Germany). Cells of all lines were grown in Minimal Eagle Medium, (MEM) (Merck SA, Darmstadt, Germany), containing 1 mmol/L L-glutamine, 0.05 mg of streptomycin, and 50 U of penicillin per 1 mL, and 10% fetal bovine serum (endogenous Zn, 0.005 mmol/L and undetectable thiamine; thiamine-deficient medium) (Merck SA, Darmstadt, Germany). The cells cultured in MEM contained TPP levels (a derivate of thiamine) 20% lower than the cells grown in MEM supplemented with 0.009 mmol/L thiamine. The number of passages before each experiment was 2 for all cell lines. Cells were seeded on 10-centimeter-diameter plates at a density of 40.000/cm^2^, and grown in MEM (thiamine-deficient medium) at 37 °C in an atmosphere of 5% CO_2_, 95% air for 48 h. At this time, the growth medium was replaced by fresh MEM experimental medium, containing Zn (Avantor Performance Materials, Gliwice, Poland), amprolium (Merck SA, Darmstadt, Germany), or both, as indicated. The culture was continued for the next 24 h. Amprolium, a competitive inhibitor of thiamine transport, was used to aggravate intracellular TPP deficiency. The complete reference medium was obtained by the addition of 0.009 mmol/L thiamine.

Culture was terminated by harvesting the cells in 10 mL of ice cold 154 mmol/L NaCl containing 5 mmol/L glucose and 10 mmol/L phosphate buffer (pH 7.4) (Avantor Performance Materials, Gliwice, Poland). Then, the cells were collected by centrifugation and suspended in 320 mmol/L sucrose with 0.10 mmol/L EDTA and 10 mmol/L Na-HEPES (pH 7.4) to obtain 10 mg/mL protein concentration. Each sample was aliquoted for Trypan Blue assay, metabolite and enzyme assays.

To determine the effect of C6 astroglial cells on Zn neurotoxicity in neurons in thiamine deficiency, highly differentiated neuronal cholinergic SN56.B5G4 [33] cells were co-cultured with C6 cells. The glial C6 cells (3 × 10^5^) were seeded on 3-centimeter-diameter inserts with a semipermeable bottom. After 24 h of preconditioning, 3 inserts were placed on each 10-centimeter plate with SN56 cells. The whole culture was continued in the thiamine-deficient medium for 24 h. Then, the amprolium, Zn or both were added as indicated, and co-culture was continued for a subsequent 24 h (Figure 2).

Primary glial cells were isolated from C57BL/6 mice (RRID: IMSR CRL: 27). The animals were taken from the Tri-City Academic Laboratory Animal Centre. They were housed in the following controlled conditions: constant temperature, 22 ± 1 °C; humidity, 50–60%; 12:12 dark–light cycle. The project was approved by the Ethics Commission on Animal Experiments at the Medical University of Gdańsk (permission no. 27/2013). No exclusion criteria were pre-determined. The animals were quickly euthanized by cervical dislocation. No procedure was used before euthanasia. Immediately afterwards, cerebral cortices were removed and mechanically dissociated in Ca^2+^ and Mg^2+^ free balanced salt solution, pH = 7.4, containing (in mmol/L) 137 NaCl, 5.36 KCl, 0.27 Na_2_HPO_4_, 1.1 NaH_2_PO_4,_ and 6.1 glucose (Avantor Performance Materials, Gliwice, Poland). The cortices were cleaned of meninges and mechanically dissociated by sequential passage through a Pasteur pipette. The pellet was suspended in a small amount of DMEM supplemented with 0.05 mg of streptomycin and 50 U of penicillin per 1 mL and 10% FBS and then counted. The cells were seeded on a 96-well plate (12 × 10^4^ cell/well) or 3-centimeter-diameter plates (10 × 10^5^), pre-treated with poly-L-lysine and grown in DMEM at 37 °C in an atmosphere of 5% CO_2_ and 95% air. The medium was changed for a fresh one every 3 days. Cells were allowed to grow to confluence and used at 21 days in vitro. This culture time was sufficient for selection of primary glial cells. Then, the medium was changed to MEM, and amprolium or Zn were added as indicated. The study was not preregistered, and no randomization was performed to allocate subjects in the study.

Blinding procedures were used as indicated: the cell culture and statistics were conducted by the experimenter; viability tests, the enzymes’ activities, acetyl-CoA, Zn, and TPP levels’ measurements were performed by different investigators working on randomized samples.

### 4.2. Viability Assays: Trypan Blue Exclusion Test, MTT Test

Equal volumes of Trypan Blue solution (Merck SA, Darmstadt, Germany) and 0.4% (*w*/*v*) isotonic were added to the 0.02-milliliter C6 and SN56 cell suspensions. The total cell number and fraction of nonviable, dye-accumulating cells were counted under a light microscope after 2 min in a Fuchs–Rosenthal hemocytometer.

The 3-(4,5-dimethylthiazol-2-yl)-2,5-diphenyltetrazolium bromide (MTT) reduction assay (Merck SA, Darmstadt, Germany) was performed in primary glial cells cultured in 96-well microplates with cytotoxic compounds as indicated for 24 h. Then, the medium was discharged and a fresh one was added with 0.60 mmol/L 3-94,5-dimethylthiazol-2-yl)-2,5-diphenyltetrazolium bromide (MTT). The incubation was continued for the next 3 h. Reduced chromophore was determined by a spectrophotometric measurement at 570 nm [34].

### 4.3. Enzyme Assays

For enzyme assays, cells were lysed and diluted to the desired protein concentration with 0.20% Triton X- 100. PDH (EC 1.2.4.1) activity was estimated using the citrate synthase coupled method. This was assayed by measurement of synthesized acetyl-CoA followed by citrate metabolism and citrate lyase. The incubation media at a final volume of 0.25 mL consisted of (in mmol/L) 50 Tris/HCl pH = 8.3, 2 MgCl_2_, 10 dithiotreitol, 2 NAD, pyruvic acid, 0.20 CoA, 2.5 oxaloacetic acid, 0.15 J.M citrate synthase, 2 thiamine pyrophosphate (Merck SA, Darmstadt, Germany), and 0.05 mL of tested sample containing 0.10 mg of protein. To measure the activity of the enzyme in situ, thiamine pyrophosphate was not added to the incubation medium. The cells were incubated with media for 30 min at 37 °C. The reaction was terminated by exposing the sample to 99 °C for 10 min. After centrifugation at 10,000 rpm, the level of citrate was measured. The media for citrate measurement at the final volume of 0.70 mL consisted of (in mmol/L) 100 Tris/HCl buffer pH = 7.4, 0.10 NADH, 0.2 J.M. malic dehydrogenase (Merck SA, Darmstadt, Germany), and 0.20 mL of tested sample. The reaction was initiated by the addition of 0.1 J.M. citrate lyase. The activity was expressed in mmols of oxidized NADH on the basis of an absorbance ratio for NADH = 6.22 mol/cm at the wavelength = 340 nm [35].

Aconitase (EC 4.2.1.3) and isocitrate dehydrogenase (EC 1.1.1.42) activities were measured by a direct measurement of NADP (Merck SA, Darmstadt, Germany) reduction. The incubation media for aconitase at the final volume of 0.80 mL consisted of (in mmol/L) 50 Tris/HCl pH = 7.4, 2 MgCl_2_, 0.10 NADP, 1 J.M. isocitrate dehydrogenase (Merck SA, Darmstadt, Germany), and 0.10 mL of tested sample containing 0.10 mg of protein. The reaction was initiated by the addition of 0.01 mL of 20 mM cis-akonitane (Merck SA, Darmstadt, Germany) [36].

The incubation media for isocitrate dehydrogenase at the final volume of 0.70 mL consisted of (in mmol/L) 50 Tris/HCl pH = 7.4, 0.60 MgCl_2_, 0.50 NADP, and 0.10 mL of tested sample containing 0.10 mg of protein. The reaction was initiated by the addition of 0.01 mL of 10 mmol/L isocitrate [37].

### 4.4. Acetyl-CoA Assay

The acetyl-CoA level was assessed after incubation of the cells in the depolarizing medium with shaking at 100 cycles/min for 30 min at 37 °C. The medium consisted, at the final volume of 1.0 mL (in mmol/L), of 2.5 pyruvic acid, 2.5 L-malate (Merck SA, Darmstadt, Germany), 90 NaCl, 30 KCl, 20 NaHEPES pH = 7.4, 1.5 Na-phosphate, 32 sucrose (Avantor Performance Materials, Gliwice, Poland), and 0.7–1.0 mg of cell protein. After incubation, the cells were separated from the medium by centrifugation for 30 s at 10,000 rpm. These conditions provided stability of the acetyl-CoA level in the cells. For total acetyl-CoA content determination, 0.20 mL of cell suspension in incubation medium was centrifuged and the pellet was deproteinized with 0.20 mL of 4% HClO_4_ (Avantor Performance Materials, Gliwice, Poland). Acetyl-CoA levels were estimated using the cycling method. Deproteinized extracts were put into a solution containing maleic anhydride (Merck SA, Darmstadt, Germany) in ethyl ether to remove CoA-SH for 2 h. The cycling reaction was carried out in 0.10 mL of medium containing (in mmol/L) 1.9 acetyl phosphate, 1.2 oxaloacetate, 1.0 IU phosphotransacetylase, and 0.12 IU citrate synthase (Merck SA, Darmstadt, Germany) for 90 min. The cycling reaction was stopped by heating the samples to 100 °C for 10 min and the citrate formed was determined [33,38]. The media for citrate measurement at the final volume of 0.70 mL consisted of (in mmol/L) 100 Tris/HCl buffer pH = 7.4, 0.10 NADH, 0.2 J.M. malic dehydrogenase, and 0.20 mL of tested sample. The reaction was initiated by the addition of 0.1 J.M. citrate lyase. The activity was expressed in mmols of oxidized NADH on the basis of absorbance ratio for NADH = 6.22 mol/cm at the wavelength = 340 nm.

### 4.5. Thiamine Pyrophosphate Assay

For TPP assay, the cells were grown on 3-centimeter-diameter plates with or without cytotoxic factors, as described in Section 2.1. After 72 h, media were withdrawn and cells that were attached to the plates were gently washed once with ice-cold PBS supplemented with 1 mmol/L EDTA to remove surface-bound metals. The collected cells were homogenized in 1 mL of ice cold 4% trichloroacetic acid. TPP was measured in supernatant by reverse-phase HPLC with an electrochemical detector [39].

### 4.6. Protein Assay

Protein level was measured using the Bradford method (1976) with human immunoglobulin as the standard.

### 4.7. Zinc Assay

Intracellular Zn level was measured in the cells cultured on 3-centimeter-diameter plates and treated as described in Section 2.1. After 72 h, media were withdrawn and the cells that were attached to the plates were gently washed once with ice cold PBS supplemented with 1 mM EDTA to remove surface-bound metals. Subsequently, 1.5 mL of ice cold 4% HClO_4_ was added on the plate and incubated on ice for 1–2 min. Then, the cells were collected into ice cold plastic tubes. Protein precipitates were removed by centrifugation for 60 s at 5000 rpm. The supernatants were neutralized with 7.5 N K_2_CO_3_ to pH 6.0. Next, the neutralized supernatants were used for Zn assay with a modified fluorometric method with N-(6-methoxy-8-quinolyl)-p-toluenesulfonamide (TSQ) (ThermoFisher Scientific, UK). The measurement was conducted in a medium that contained 1.60 mL of 0.10 mmol/L Na-HEPES pH 7.4, 0.40 mL of neutralized supernatant, and 0.02 mL of TSQ in a 2-milliliter quartz cuvette. The analysis was conducted on a fluorescence spectrometer (LS-55, Perkin Elmer) with emission and excitation wavelengths equal to 335 and 495 nm, respectively, against a blank reagent that was treated similarly.

To quantify analysis, the peak of emission was calculated from the calibration curve in the range 0.2–4 nmols/sample of ZnCl_2_ standard solution treated similarly to the samples [40].

### 4.8. Statistical Analyses

Statistical analyses were carried out using GraphPad Prism 5 (GraphPad software, La Jolla, CA, USA). No sample calculation was performed. Data are presented as means ± SEM or median ± Whiskers: Min to Max from 3–6 independent cell culture preparations. The assessment of normality of the data was carried out using the Shapiro–Wilk normality test. Data analysis was carried out using the Kruskal–Wallis normality test. The two groups were compared using a non-parametric Mann–Whitney test, with *p* < 0.05 being considered to be statistically significant. No test for outliers was used.

## 5. Conclusions

Although the development of medicine has made our lifespans longer, there is a higher risk of the progression of neurodegenerative disease. It is estimated that 35 million people suffer from Alzheimer’s disease worldwide. Moreover, it is predicted that this number will triple by 2050. AD is also compounded by thiamine deficiency, especially among elderly people who are malnourished. There is no effective therapy against any neurodegenerative disease. Therefore, knowledge about the molecular mechanism of AD should help in the development of a successful therapy. The presented data indicate that astroglial cells are more resistant to thiamine deficiency and Zn excess than neuronal cells. This may be caused by similar levels of intracellular Zn accumulated in TD-C6, causing a much weaker inhibition of PDH, aconitase and ICDH activities, and acetyl-CoA levels than in neuronal TD-SN56 cells. Thanks to this, in a co-culture, astroglia may retain their ability to take up Zn excess and provide TPP to the neuronal vicinity. Thus, astroglia could prevent the inhibition of PDH and aconitase and ICDH-NADP activities and maintain the acetyl-CoA needed for neuronal viability. Therefore, therapeutic approaches that protect astrocytes against cytotoxic insults may ensure preservation against zinc overload in cholinergic neurons.

## Data Availability

The data that support the findings of this study are available from the corresponding author upon reasonable request.

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
