# Peer review of "Protection of Cholinergic Neurons against Zinc Toxicity by Glial Cells in Thiamine-Deficient Media"

_ijms, 2021, doi:10.3390/ijms222413337_

Round 1

Reviewer 1 Report

Dear authors,

Thank you for submitting your manuscript for peer review. I understand in your research outlined in this manuscript, you demonstrate that in an in vitro co-culture system combining neurons and glial cells, glial cells protect the neurons against cytotoxicity mediated by Zn and thiamine deficiency. 

The work presented in the manuscript is significant and interesting. However, the presented text reads like a draft and is difficult to understand easily. The manuscript must be written and presented using a high standard of academic English for peer-review. Peer-reviewers are not to be expected to present a list of the English flaws in the text. I understand English is the primary language of the authors and I commend them for presenting their work in English. However, I do encourage the authors to seek assistance from an academic English editor or an editing company before submitting their work for peer-review. I will be in a better position with a clearly written text to provide a fair peer-review. I apologize that I cannot be positively critical in this case. Thank you for the opportunity. 

I have attached the PDF version of the original manuscript with some highlighted sections that have been distracting and need revision. These are examples only. 

Author Response

Dear Editor and Reviewers,

Thank you for reviewing our manuscript entitled: Mutual toxicity of marginal Zn excess  in thiamine deficient neuronal and glial cells by Gul-Hinc et al (ijms-1441871).

According to your suggestions, we decided to undertake a major reworking of the paper prior to resubmission. The reviewers provided insightful comments and suggestions to thoroughly revised the manuscript to fully address the criticism.

Considering the Reviewers’ suggestions, almost all sections have been revised. Also a professional English proof was undertaken Specialist Translation Agency Atominum. Moreover the title of the manuscript was changed.

We hope that the new version has better text fluidity and fulfil the Reviewers’ expectations.

Reviewer 2 Report

This manuscript is a nice study by Gul-Hinc et al, where the authors attempted to identify the toxic effects of excess Zn ion in thiamine pyrophosphate deficient neuronal and glial cells. I like the simple study design, yet an interesting outcome of this study. Hence, this could be a nice addition to MDPI-IJMS. However, I have a couple of comments below. If the other reviews are favorable, I would like the authors to explain this before this can be published.

Comments:

(1) My major concern is that this manuscript is very hard to read. I agree that the subject matter is complicated, that contains knowledge from a very specific organometallics field. But at the same time, I don’t think the English writing is easy to follow. I am giving a couple of examples, right at the beginning of the abstract. The authors tried to convey some message, that can definitely be written in better English:

“Exposition (Not a right word, should be “exposure”) of neuronal SN56 cholinergic cells to combined sub-toxic zinc excess and mild thiamine deficits aggravated their injury due to interference with acetyl-CoA metabolism. The aim of this work was (missing to) investigate whether and how in such conditions astroglial C6 cells might alleviate neurotoxicity in co-cultured SN56 cells.”

I also don’t like the title, it’s too complicated to understand and someone has to thoroughly read the actual manuscript in details to get the message. The authors should use a much simpler title to get the readers oriented towards the topic already at the beginning.

I think, before resubmitting, the authors should give the revised manuscript to few people around to make sure that their message is rightly conveyed.   

(2) The authors should also describe and comment about the future impacts of the study with further details with the prospective clinical benefits. Right now, I agree that they have few things about this in the introduction, but that is rather vague. They should highlight few key studies in this field and impress on the impact of their results in a point-by-point format in the conclusion section. This would help readers fully comprehend the subject after they see the results from the authors.   

Author Response

(The authors gave the same response as above.)

Round 2

Reviewer 1 Report

The text of the manuscript has been improved since the initial submission. Some language issues still remain, but these are minor and potentially can be corrected at the typesetting stage. I have some minor suggestions, which I hope will help the authors improve the text. I list these below in no specific order.

  1. The word ‘from’ should be changed to ‘against’ in the title. It seems the authors have problem with using the correct prepositions. In the title, change ‘in cases of thiamine deficiency’ to ‘in thiamine-deficient media’. In line 11, please change ‘in mild’ to ‘under mild’. In line 21, please change ‘from’ to ‘against’. Use the same convention throughout the manuscript.
  2. Avoid noun trains, long strings of nouns acting as adjective. For example, see line 22, ‘combined Zn and thiamine deficiency neurotoxic signals’.
  3. Despite being the commonly used and commonly accepted ‘misnomers’, the eponymous terms ‘Parkinson’s Disease’ or ‘Alzheimer’s Disease’ (AD) are logically incorrect. Historically, the diseases were discovered by Charles Parkinson or Alois Alzheimer, respectively; the diseases were not ‘their own’ diseases. Because of the eponymous convention, using the possessive form (apostrophe plus ‘s’ or genitive ‘s’) is wrong but has been perpetuated in the English Scientific literature by our great peers. The Australian Manual of Scientific Style and The Chicago Manual of Style also advise against the use of the possessive form. I suggest taking their editorial advice and applying it throughout the text. Many authors have avoided this, for example, ‎Arispe N, Pollard HB, Rojas E. Giant multilevel cation channels formed by Alzheimer disease ‎amyloid β-protein [AβP-(1–40)] in bilayer membranes. Proc Natl Acad Sci USA. 1993;90(22):10573-7.‎ Arispe N, Pollard HB, Rojas E. β-Amyloid Ca(2+)-channel hypothesis for neuronal death in ‎Alzheimer disease. Mol Cell Biochem. 1994;140(2):119-25.‎ Arispe N, Rojas E, Pollard HB. Alzheimer disease amyloid β protein forms calcium channels ‎in bilayer membranes: blockade by tromethamine and aluminum. Proc Natl Acad Sci USA. ‎‎1993;90(2):567-71.‎ Serrano-Pozo A, Frosch MP, Masliah E, et al. Neuropathological alterations in Alzheimer ‎ Cold Spring Harb Perspect Med. 2011;1(1):a006189.‎ Bruno MA, Leon WC, Fragoso G, et al. Amyloid β-induced nerve growth factor ‎dysmetabolism in Alzheimer disease. J Neuropathol Exp Neurol. 2009;68(8):857-69. ‎Use ‘Alzheimer disease’ instead of the genitive form.
  4. Note the WHO-accepted nomenclature for ‘amyloid β-protein’ and use this nomenclature throughout the manuscript instead of ‘amyloid-β; the abbreviated form as Aβ is OK.
  5. Please present a schematic of the biochemical pathways and reactions relevant to this study and refer to that schematic in the introduction of the manuscript. This is in reference to lines 66–84 and other relevant parts of the text.
  6. The labels in many of the figures are in small font. Please revise the figures and the corresponding labels.
  7. The interpretation in line 114–116 seems is confusing because figure 1b clearly shows a difference between the two conditions. Please revise all the conclusions of the results to ensure correct representations of the results.
  8. The results mentioned in line 136 are not shown. Please specify this if my understanding is correct.

Author Response

Thank you for reviewing our manuscript entitled: Mutual toxicity of marginal Zn excess in thiamine deficient neuronal and glial cells by Gul-Hinc et al (ijms-1441871).

According to your suggestions, we have made all necessary changes prior to resubmission.

We followed all of the Reviewers’ suggestions, as listed below.

  1. The suggested changes in the title:

a)‘from’ has been changed to ‘against’;

  1. b) ‘in cases of thiamine deficiency’ to ‘in thiamine-deficient media’
  2. In line 11, the term ‘in mild’ has been changed to ‘under mild’.
  3. In line 21, the preposition ‘from’ has been changed to ‘against’. The same convention is undertaken throughout the manuscript.
  4. The long strings of nouns acting as adjective are revised throughout the manuscript.
  5. The genitive form :”Alzheimer’s Disease” has been changed to “Alzheimer disease” throughout the manuscript.
  6. According to WHO-accepted nomenclature: ‘amyloid β-protein’ has been used instead of ‘amyloid-β’.
  7. We have made a schematic figure and have placed it in the Introduction chapter.
  8. The labels of the figures have been enlarged.
  9. The interpretation of figure 1b has been changed according to the differences in TPP content in thiamine supplemented or deficient medium.
  10. The results of total C6 cell number under 0.30 mmol/L Zn (line 136) have been added to the figure 2b.

We hope that the new version fulfil the Reviewers’ expectations.